# Elbow Stiffness Imaging: A Practical Diagnostic and Pretherapeutic Approach

**DOI:** 10.3390/jcm10225348

**Published:** 2021-11-17

**Authors:** Charles Lombard, Pedro Teixeira, Edouard Germain, Gauthier Dodin, Mathias Louis, Alain Blum, Romain Gillet

**Affiliations:** Guilloz Imaging Department, Central Hospital, University Hospital Center of Nancy, 54035 Nancy, France; c.lombard@chru-nancy.fr (C.L.); p.teixeira@chru-nancy.fr (P.T.); e.germain@chru-nancy.fr (E.G.); gauthier-dodin@orange.fr (G.D.); m.louis@chru-nancy.fr (M.L.); alain.blum@gmail.com (A.B.)

**Keywords:** elbow stiffness, MRI, CT, 4D-CT, elbow osteoarthritis

## Abstract

Loss of elbow motion can lead to disability in everyday gestures, recreational activities, and work. Unfortunately, the elbow joint is particularly prone to stiffness because of its complex anatomy and biomechanics. The etiology of elbow stiffness is varied and must be diagnosed accurately in order to allow optimal treatment, which may be challenging for surgeons and physiotherapists. Its treatment can be either conservative, arthroscopic or surgical, with a trend for arthroscopic procedures when conservative treatment fails. There is no consensus on the optimal imaging workup for elbow joint stiffness, which may have an impact on patient management. This article reviews the current classification systems of elbow stiffness and the various imaging techniques used for diagnosis. Report checklists and clarifications on the role of each imaging method, as well as the imaging findings of normal and stiff elbows, are presented, leading to a proposed diagnostic algorithm. The main concern in imaging is to determine the cause of elbow stiffness, as many concomitant abnormalities might be present depending on the clinical scenario.

## 1. Introduction

The elbow joint is a complex hinge-type synovial joint with an important role in the mobilization of the upper limb, linking the hand, wrist, and shoulder. The elbow allows precise hand positioning and serves as a forearm fulcrum maximizing grip strength [1]. Loss of elbow motion can lead to disability in everyday gestures, recreational activities, or work [1]. Unfortunately, the elbow joint is particularly prone to stiffness because of its complex anatomy and biomechanics [2,3]. Post-traumatic changes in the elbow’s peri-articular tissues predispose to capsular calcification and ossification [4,5,6]. Fractures around the elbow, even if non-displaced and adequately treated, may require sustained immobilization due to difficulties in obtaining a stable osteosynthesis, which may also contribute to joint stiffness [6]. Histologically, elbow joint stiffness is thought to result from post-traumatic capsular thickening with disorganized collagen fibers, altered cytokine levels, and elevated myofibroblasts [7].

The etiology of elbow stiffness is varied and must be diagnosed accurately in order to allow the best therapeutic management, which may represent a challenge for surgeons and physiotherapists. Its treatment can be either conservative (e.g., physiotherapy and splinting), arthroscopic (e.g., most frequently: anterior capsular resection, cleansing the humeral fossae, osteophyte and loose bodies ablation) or surgical (e.g., open elbow arthrolysis and prosthetic joint replacement) [8,9]. Although a well-defined therapeutic algorithm has not yet been proposed, the frequency of postsurgical complications has led to favor arthroscopic procedures when conservative treatment fails [8,9].

There is also no consensus on the optimal imaging workup for elbow joint stiffness, which may have an impact on patient management. Radiographs and CT scan allow a good analysis of osseous structures and joint spaces, while CT arthrography improves loose body detection and cartilage assessment. Although MRI excels in demonstrating capsular and periarticular soft tissue anomalies, its role in elbow stiffness workup is not well defined. In this article, after a key anatomical and biomechanical review, the different etiologies of elbow stiffness are reviewed, and a practical diagnostic and pretherapeutic imaging approach is proposed.

## 2. Anatomy and Biomechanics

The elbow encompasses three joints within a single articular capsule: humeroradial, humeroulnar, and proximal radioulnar. The distal humerus presents two condyles: the trochlea, medially, which articulates with the greater sigmoid notch of the proximal ulna, and the capitellum, laterally, which articulates with the radial head. The trochlea and the capitellum are anteverted by 30°, and they have a 5° medial rotation and a 6° valgus with respect to the humeral long axis. Anteriorly, radial and coronoid fossae lodge the radial head and coronoid process, respectively, during elbow flexion. Posteriorly, the olecranon fossa lodges the olecranon during extension. The radial head articulates medially with the radial notch of the ulna allowing forearm pronosupination.

The two humeral epicondyles harbor the insertions of various ligaments and tendons. The ulnar collateral ligaments and flexor–pronator tendon group insert onto the more prominent medial epicondyle while the lateral collateral ligament and extensor–supinator tendon group insert onto the less prominent lateral epicondyle [1].

Stability, mobility, and alignment are essential prerequisites for elbow function [3]. The maximal elbow flexion–extension range is from 0° to 150° with 75° forearm pronation and an 85° supination [1]. However, the minimal functional range of motion needed for daily living tasks is 30–130° of flexion–extension with 50° of pronosupination [10]. Tasks such as using a cell phone or a keyboard require at least 142° of flexion–extension and 65° of pronation [11]. Elbow stiffness is defined by a flexion–extension range from 30 to 120° or a forearm pronosupination inferior to 45°. Stiffness in flexion is less tolerated than in extension. Loss of supination is more devastating than the loss of pronation, which may be partially compensated by shoulder abduction [4,12]. Patient activity should also be taken into account, as athletes, musicians, and manual workers may require a greater superior limb range of motion than the general population. 

The two main mechanisms that contribute to elbow stiffness are blocks, corresponding to compressive resistance in the direction of the motion, and tethers, corresponding to tensile resistance in the opposite direction of the motion [13]. Anterior tethers and posterior blocks can cause extension deficit, while posterior tethers and anterior blocks may lead to a flexion deficit (Table 1). According to Sun et al., tethers can be found alone, but blocks are often associated with tethers [13].

## 3. Clinical Presentation

Acute or repeated trauma remains the most frequent cause of elbow stiffness, followed by osteoarthritis (OA). Elbow pain is usually mechanical in origin and appears in extreme degrees of motion limitation. Spontaneous elbow pain is unusual. In such cases, an infection should be considered and lead to prompt patient management (e.g., blood work, articular puncture/lavage, and antibiotics if septic arthritis is confirmed).

Several classifications of elbow stiffness have been proposed according to the structures involved, anatomic location, mechanism of injury, or severity of motion loss [2,3]. One of the most relevant classification systems from an imaging and clinical perspective is the one described by Morey et al., which divides elbow pathology into extra-articular; intra-articular, or mixed (the most frequent pattern) [4]. More recently, Sun et al. proposed a motion-based classification system. Elbow flexion–extension dysfunction is divided into four categories—tethers alone, tethers with blocks, articular malformation, or bony ankyloses, while forearm pronosupination dysfunction is divided into three—contracture alone, radial head malunion/nonunion, or proximal radioulnar bony ankyloses [13].

### 3.1. Extra-Articular Elbow Stiffness

Extra-articular elbow stiffness may be related to periarticular tissue pathology (e.g., articular capsule, muscles, ligaments, and skin), heterotopic ossifications, extra-articular bone malalignment or a combination of these. Extra-articular stiffness is frequently posttraumatic, in particular dislocations and complex elbow fractures, although simple nondisplaced radial head fracture or elbow subluxation could lead to stiffness especially after prolonged immobilization. Increased cast immobilization time, alcohol abuse, and prior joint surgery are also considered risk factors [12].

The diagnosis of skin involvement (usually treated by skin plasty), whether it is a large scar or burn, is clinical and does not necessarily require imaging, unless other associated lesions are suspected, in particular heterotopic ossifications (HO), in the setting of neurogenic paraosteopathy [13,14]. The latter consists of the formation of mature bone lamellae in soft tissues, which should be differentiated from capsular or ligamentous calcifications or ossifications (Figure 1). Periarticular HO occurs after direct elbow traumatism (up to 3% in simple dislocation and 20% in fracture–dislocation) and may affect elbow flexion–extension but also lead to a forearm pronosupination deficit due to radioulnar synostosis formation [15] (Figure 2). HO risk factors include local burn, concomitant head or spinal injury, prolonged immobilization, and delays before surgery [16]. There is no consensus on HO treatment, which can be conservative or surgical. Botulinum toxin injections have been shown to be efficient [17]. Radiation therapy has also been proposed for HO prevention [16,17]. Joint capsule contractures can be secondary to trauma, arthritis (whether inflammatory, septic, or secondary to repetitive hemarthrosis in hemophilia), and surgery. The anterior capsule seems to be more frequently thickened than the posterior, explaining the preferential loss of extension rather than flexion (Figure 3). Other soft tissues such as muscle and ligaments can also be involved. Finally, bone malalignment can lead to stiffness especially in cases of extra-articular elbow fracture malunion or congenital anomalies (e.g., congenital dislocation of the radial head or arthrogryposis). Treatment is usually surgical [13].

### 3.2. Intra-Articular Stiffness

The mechanisms of Intra-articular stiffness are multiple and can be combined. The most frequent are chondropathy (whether of posttraumatic origin, related to osteochondritis or as part of an OA), primary or secondary synovial chondromatosis, posttraumatic joint surface incongruence, and intra-articular adhesions. Proximal radioulnar joint arthritis, an incongruent radial head (Figure 4), or adhesions between the radial head and the annular ligament may lead to pronosupination deficits.

OA remains one of the main causes of intra-articular stiffness. It is more often secondary to a traumatic injury, regardless of its severity; however, it can rarely be primitive, especially in manual laborers, weight lifters, and throwing athletes [18]. Distal humeral fracture and elbow fracture–dislocations are more prone to lead to OA than olecranon or radial head fractures [19]. In OA, the first mechanism of stiffness is mechanical impingement related to osteophytes at the extremes of flexion and extension (Figure 5) rather than cartilage surface damage. These osteophytes typically appear in the early stages of osteoarthritis and will classically develop on the tips of the olecranon and the coronoid or fill the coronoid and olecranon fossa (Figure 6) [5]. As for extra-articular causes, treatment is usually surgical or arthroscopic [13]. A total elbow arthroplasty is an option for joint ankyloses or advanced arthropathy [20].

## 4. Imaging Assessment of Elbow Stiffness

The aim of imaging is threefold: confirm the etiology of stiffness, determine which anatomic elements are involved, and allow optimal treatment planning (procedures and arthroscopic or surgical approaches). Therefore, the following elements should be evaluated in the standard imaging workup:

Remaining bone stock;

Bone alignment;

Articular congruency;

OA location and severity;

Presence of osteophytes: presence and repercussions (impact on articular mobility and relation to neurovascular structures);

Loose bodies (embedded in the synovial membrane or free);

Presence of surgical implants (articular protrusion?);

Soft tissue pathology.

## 5. Imaging Workup

### 5.1. Conventional Radiographs

The initial assessment of a stiff elbow should always begin with standard radiographs, with at least anteroposterior (AP) and lateral views. For the AP view, the patient is positioned with the elbow in full extension and the forearm in supination. It allows optimal visualization of the medial and lateral epicondyles, the radiocapitellar joint, and the coronal bones alignment. In case of inability to fully extend the elbow due to contracture, two anteroposterior views should be performed—one perpendicular to the distal humerus and one perpendicular to the proximal radius and ulna. CT is an option if those views cannot be obtained. For the lateral view, the patient is positioned with the elbow in 90° flexion and the forearm in a neutral position, allowing a good assessment of the humeroulnar joint, coronoid, and olecranon. Humeral anteversion (30°) and articular congruency can also be checked with this view.

Additional oblique views can be performed according to the clinical suspicion. The medial oblique view consists of an AP view with a 45° medial rotation of the arm and forearm, improving the visualization of the trochlea, olecranon, and coronoid processes. The lateral oblique view is obtained with a 45° lateral rotation of the arm and forearm, improving the visualization of the radiocapitellar joint, medial epicondyle, radioulnar joint, and coronoid process. The radial head view consists of a lateral view with a 45° cranial angulation of the incident beam with respect to the humerus, allowing better visualization of the radial head and radiocapitellar joint. CT is gradually replacing these complementary radiographic views, which may be an advantage in the author’s opinion. 

One must keep in mind that non-ossified articular loose bodies (i.e., cartilaginous) and soft tissues cannot be seen on standard radiographs and that filling of the olecranon and coronoid fossa can indicate the presence of osteophytes, which are better depicted on CT (Table 2).

### 5.2. Ultrasonography

Ultrasonography can be recommended for the evaluation of ligamentous, tendinous, nerve pathologies, and synovitis (Table 3). It is not the method of choice for the evaluation of loose bodies and osteochondral traumatic lesions and should only be used for this purpose when other imaging methods are not available [22]. The ulnar nerve deserves special attention, as it is often involved in traumatic or degenerative processes. Additionally, a “stretching” neuropathy can occur after treatment when the patient recovers normal motion. Radial and median nerves are less susceptible to injury but can be affected by scar formation, heterotopic ossification, or iatrogenic injury [3]. As their pathology can lead to instability, collateral ligaments should be checked if signs of instability are clinically present [23].

### 5.3. CT and CT Arthrography

CT is superior to radiographs in identifying and characterizing the osseous causes of elbow stiffness [24]. It allows a better assessment of osteophytes, which are almost always present in OA patients at the trochleo-ulnar compartment (e.g., anterior coronoid area and medial part of the trochleo-ulnar joint) and in about 25% of the cases at the radiocapitellar compartment (e.g., both at the anterior and posterior portions). It also allows a precise joint space delineation, which permits the diagnostic of severe OA, defined by any degree of joint space narrowing by Kwak et al. [25]. Late elbow OA has been shown to present worse clinical and radiologic outcomes when treated by arthroscopic osteocapsular arthroplasty with respect to early OA. Severe OA is also a contraindication to isolated contracture release procedures, and in these cases, elbow arthroplasty should be considered [7]. The number and location of ossified intra-articular osteochondromas, the location and anatomic relations of HO (including their relations to neurovascular structures), and also study of complex fractures are facilitated by CT. A 3D volume rendering or preferably global illumination reconstructions should be performed to provide the surgeon with the cartography and relations of the ossified pathologic processes necessary for optimal surgical planning. 

CT-arthrography is minimally invasive and allows an optimal evaluation of the cartilage articular surface (Figure 7), and is recommended prior to treatment of elbow osteoarthritis [26]. If such cartilaginous lesions are present, osteophytes treatment alone may not totally improve elbow function, and pain may remain after surgical management [5]. Outcomes after arthroscopic treatment are better when no cartilaginous lesion is observed, but current classification systems cannot be used as prognostic factors before treatment [5]. Finally, signs of impingement (osteophytes and filling of fossa) are more common than cartilaginous lesions and could be considered as a pre-arthritic stage.

CT arthrography should be preceded by a non-contrast acquisition to detect calcified loose bodies that by be obscured by the intra-articular iodinated contrast. As regards the evaluation of intra-articular osteochondromas, CT has two main purposes—location and mobility assessment (e.g., mobile after arthrography or not) (Figure 8). CT arthrography is particularly useful for the identification of non-ossified intra-articular loose bodies (e.g., cartilage fragments). Patients with loose bodies and no cartilaginous lesions on CT-arthrography have been shown to be good candidates for arthroscopic treatment of OA [26]. Moreover, CT-arthrography can also lead to capsular retraction diagnosis if the articular capacity is reduced to around 6 ± 3 mL. An intra-articular corticosteroid injection can be coupled with CT arthrography, serving as a diagnostic and therapeutic test (e.g., reduction of capsular inflammation).

A CT and CT-arthrography checklist is presented in Table 4.

### 5.4. MR Imaging

MRI has a limited role in the assessment of elbow stiffness and may not be required [27]. Its main interest is soft-tissue evaluation, especially searching for capsular and/or ligament thickening, as the loss of soft tissue elasticity is thought to be the result of bleeding, edema, granulation tissue formation, and fibrosis, which may translate to a low signal intensity (e.g., in both T1 and T2 weighted images) capsular thickening on MRI (Figure 3). Depending on the amount of joint fluid, intra-articular adherences can also be seen as tissue bands connecting the joint capsule to osseous structures. Non-fat-saturated T1 weighted sequences are also useful to identify fibrotic changes in the intra-articular fat-pads (e.g., low signal intensity bands and peripheral thickening). MRI should be performed when CT arthrography does not provide a clear explanation for joint stiffness or when soft tissue involvement is suspected (e.g., synovitis, ligament dysfunction, neuropathies) [13]. Moreover, intra-articular space-occupying lesions such as cystic ganglia may not be identified on CT and may cause neuropathies or induce elbow stiffness when located at the anterior joint compartment [28]. Neuropathies around the elbow constitute another MRI indication. For instance, the ulnar nerve is superficially located and susceptible to trauma and impingement due to degenerative joint changes [29]. Unexplained atraumatic elbow contracture with negative radiographs may be at times related to elbow benign and malignant soft tissues tumors that should be evaluated on MRI [30] (Figure 9). A particular situation should be noted: when the patient’s primary complaint after trauma is pain rather than stiffness without osseous lesions on radiographs or CT, type 1 complex regional pain syndrome should be considered and ruled out, as those patients may not respond to conventional therapeutic strategies [7]. MR arthrography is also an option, allowing a combined evaluation of capsular and periarticular soft tissues, cartilage articular surface, and intra-articular loose bodies. 

An MRI diagnostic checklist is presented in Table 5.

### 5.5. Dynamic and Kinematic CT

Dynamic CT should be performed in the preoperative evaluation of patients with elbow stiffness caused by bone lesions, intra-articular loose bodies, and periarticular ossifications/calcifications, particularly when multiple anomalies are present. Indeed, multifactorial joint blocks are frequent in clinical practice, and identifying the origin of the impingement with static imaging methods can be difficult (Figure 10). In this context, step-and-shoot acquisitions in maximal flexion and maximal extension may be helpful when dynamic CT is not available. Similarly, acquisitions in full pronation, neutral position, and full supination can be used for the evaluation of pronosupination deficits.

Wide-area detector scanner models allow the acquisition of kinematic 4-D datasets of the elbow during motion, by repeating low-dose acquisitions for about 7–8 s with a high temporal resolution (e.g., inter volume delays as low as 0.27 s). This method has been used for the evaluation of dynamic pathologic processes in various joints and can be used for the evaluation of the elbow during flexion–extension and pronosupination maneuvers [29,30,31,32,33,34,35]. Elbow kinematic CT can be seen as a problem-solving tool recommended for the evaluation of bony impingement when acquisitions in the extremes of joint position are not sufficient to clearly determine the nature of the impingement. 

### 5.6. Rationale for Determining the Optimal Imaging Workup and Diagnostic Algorithm

Medical history, physical examination, and initial imaging workup with conventional radiography and ultrasonography usually allows differentiating between intra- or extra-articular origin of elbow stiffness but are rarely sufficient if surgical treatment is being considered (arthrolysis, interposition arthroplasties, and prosthetic replacements) [6]. Non-surgical treatment would be more effective in case of the absence of advanced joint derangements usually identified on conventional radiographs and ultrasound [6]. Additionally, both Kay’s and Morreys’ classification systems include soft tissue and bony anatomy, underscoring the need for a proper analysis of both articular and periarticular anatomy, which may require CT-arthrography or MRI. Thus, based on the presented information a diagnostic algorithm is proposed (Figure 11).

## 6. Conclusions

Although the etiology of elbow stiffness is multifactorial, it is most frequently caused by an association of tethers and/or blocks, essentially secondary to trauma and OA. Elbow stiffness can involve multiple intra-articular or extra-articular structures, such as capsular, and periarticular soft tissues and the imaging workup is paramount for optimal patient management. In addition to medical history and physical examination, radiographs and CT (ideally with acquisitions in the extremes of joint motion) represent the cornerstone of the imaging workup, illustrating and characterizing bony impingement. CT-arthrography is an effective tool in the preoperative setting allowing an optimal evaluation of the cartilage articular surface. Finally, MRI can be recommended when conventional radiographs and CT are inconclusive, especially if patients with atraumatic joint stiffness searching for a soft-tissue origin for joint stiffness.

## Figures and Tables

**Figure 1 jcm-10-05348-f001:**
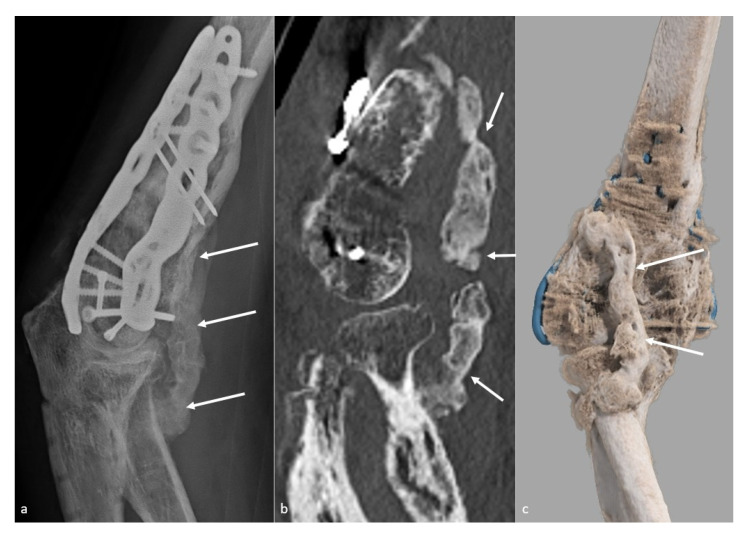
**Heterotopic ossification forming a bony bridge between the humerus and the radial neck**. Heterotopic ossification (white arrow), leading to severe loss of flexion in a 54-year-old man who suffered from a complex fracture–dislocation, shown (**a**) on a profile view radiograph, (**b**) a sagittal CT-scan reformat, and (**c**) a global illumination 3D reformat.

**Figure 2 jcm-10-05348-f002:**
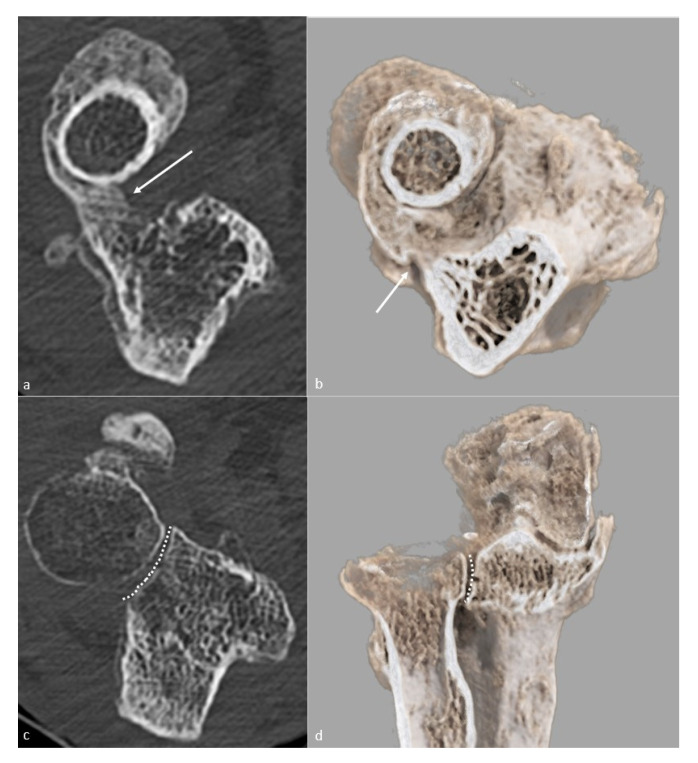
**Proximal radioulnar synostosis**. Proximal radioulnar synostosis (white arrow), leading to complete loss of forearm rotation in the same patient than in Figure 1, is shown on (**a**) an axial CT-scan view and (**b**) a global illumination 3D reformat. Note that the proximal radioulnar joint space (dotted line) can be considered normal and is shown (**c**) on an axial CT-scan view and (**d**) a global illumination 3D reformat.

**Figure 3 jcm-10-05348-f003:**
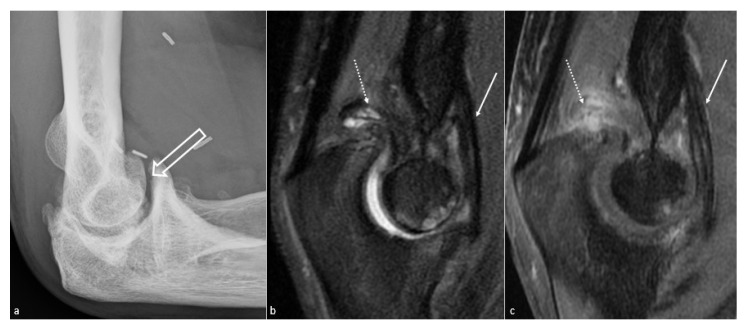
**Anterior capsular thickening**. (**a**) Profile view radiograph shows posttraumatic humeroulnar arthrosis (big white arrow), in a 45-year-old woman who suffered from a humeral fracture and presented an extension deficit. (**b**) Sagittal fat saturated T2-weighted images and (**c**) fat saturated T1-weighted gadolinium-enhanced images show anterior capsular fibrous thickening (white arrow). Additionally, note the posterior recess synovitis (dotted arrow), with a notable enhancement in (**c**), whereas no capsular thickening is seen, which also participates in the extension deficit.

**Figure 4 jcm-10-05348-f004:**
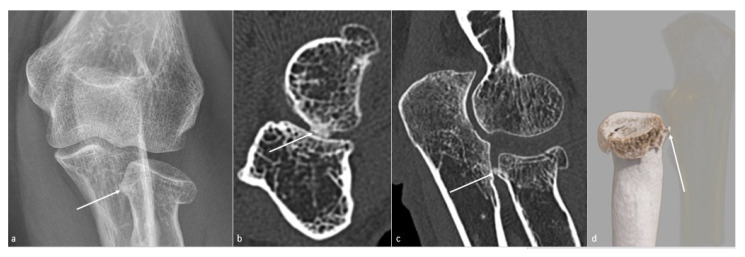
**Posttraumatic radial head vicious bone callus.** A vicious posttraumatic radial head bone callus (white arrow), causing forearm rotation dysfunction in a 47-year-old woman, is poorly defined on (**a**) an anteroposterior radiograph. The callus and its particular location are better seen on (**b**) an axial and (**c**) sagittal CT-scan reformat, and clearly defined on (**d**) a global illumination 3D reformat (ulna is voluntarily shown transparent).

**Figure 5 jcm-10-05348-f005:**
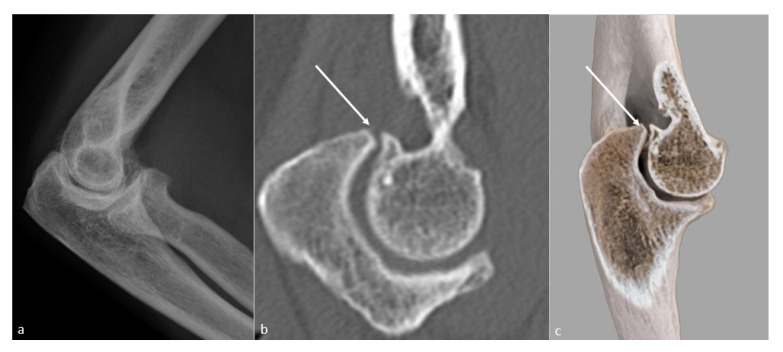
**Loss of extension secondary to a bony osteophytic impingement.** A bone block secondary to a trochlear osteophyte (white arrow) in a 51-year-old man, manual worker, is shown on (**a**) a profile view radiograph (not depicted), (**b**) a sagittal CT-scan reformat, and (**c**) a global illumination 3D reformat.

**Figure 6 jcm-10-05348-f006:**
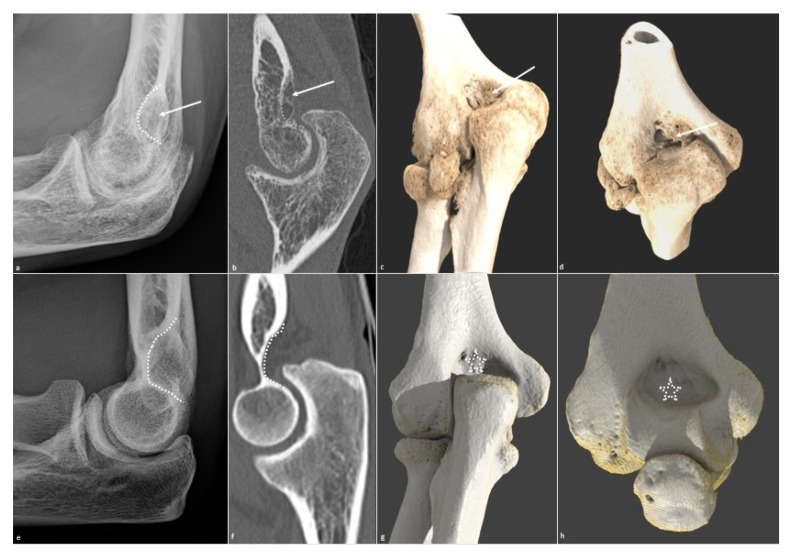
Olecranon fossa osteophytic filling. Osteophytic filling (white arrow) of the olecranon fossa (white dotted line) is shown on (**a**) a profile radiograph and (**b**) a sagittal CT-scan in a 26-year-old man who suffered from a humeral fracture and presents extension dysfunction when its corresponding normal aspect is shown in (**e**,**f**). The filling is also seen on (**c**) an extension global illumination 3D reformat and (**d**) in flexion (white arrows), when the normal aspect is shown in (**g**,**h**) (dotted white star).

**Figure 7 jcm-10-05348-f007:**
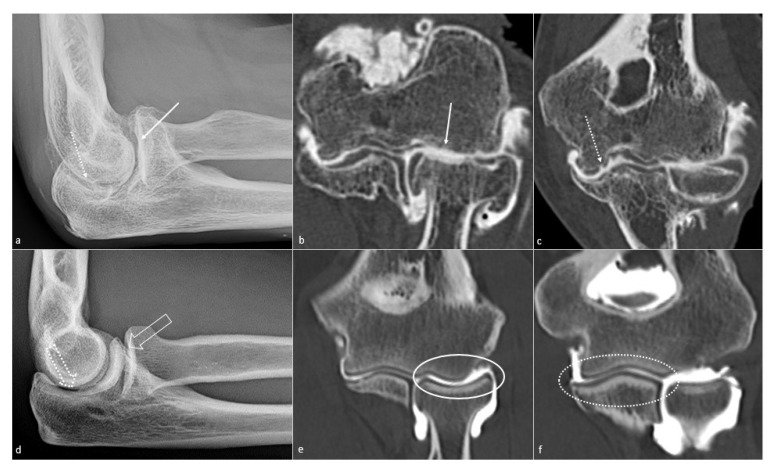
**Elbow osteoarthritis.** Elbow osteoarthritis in a 57-year-old man suffering from posttraumatic stiffness since she was 12 years old is seen on (**a**) a profile radiograph, with humeroradial (white arrow) and humeroulnar (dotted white arrow) joint space narrowing; humeroradial chondrolysis is depicted on (**b**) a frontal CT-arthrography view (white arrow), and humeroulnar cartilaginous erosions are shown on (**c**) a frontal CT-arthrography view (dotted white arrow). Normal corresponding aspects are shown in (**d**) a profile radiograph, (**e**) a frontal CT-arthrography view of the humeroradial joint (white circle) and (**f**) a frontal CT-arthrography view of the humeroulnar joint (dotted white circle).

**Figure 8 jcm-10-05348-f008:**
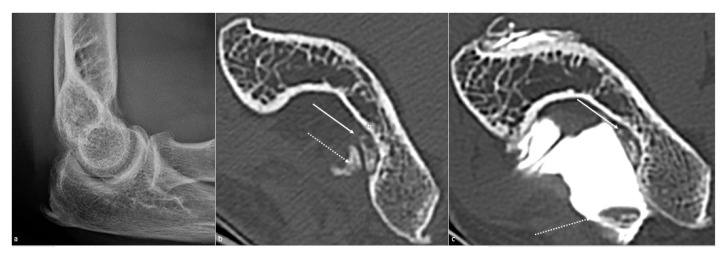
**Articular loose bodies.** Olecranon loose bodies are shown, not depicted on (**a**) a profile radiograph but well seen on (**b**) an axial CT-scan view (arrows). (**c**) CT arthrography allows identifying a free lose body (dotted white arrow) and a synovial-embedded one (white arrow), in a 57-year-old woman presenting osteoarthritis secondary to a radial head fracture.

**Figure 9 jcm-10-05348-f009:**
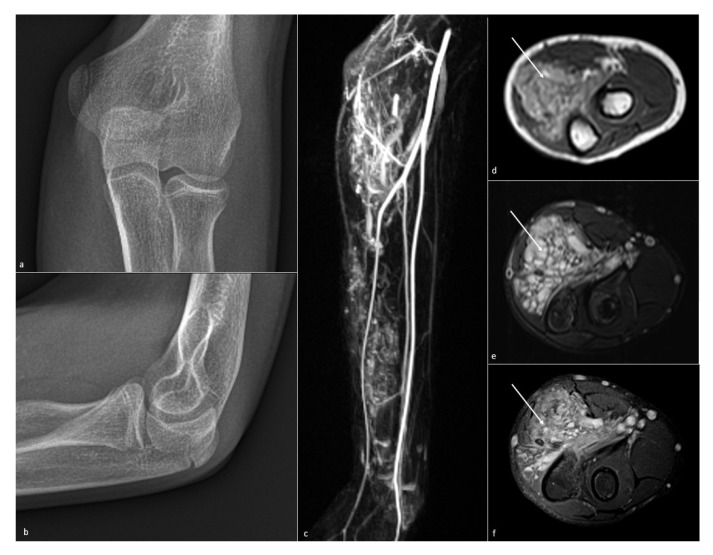
**Vascular malformation of the forearm.** Vascular malformation of the forearm causing elbow stiffness in a 12-year-old boy. (**a**,**b**) No phleboliths or bony anomalies are visible on radiographs. (**c**) Angio-MR sequence, (**d**) axial T1-weighted, (**e**) T2-weighted fat-saturated, and (**f**) T1-weighted gadolinium-enhanced fat-saturated show a fatty and heterogeneous lesion (white arrow), with high vascular enhancement, in the anterior forearm compartment.

**Figure 10 jcm-10-05348-f010:**
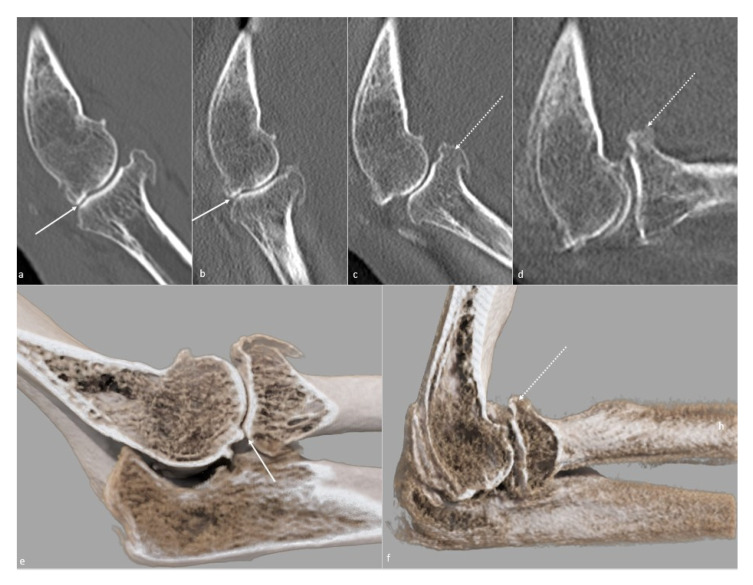
**Osteophytic bone block on 4D cinematic CT.** (**a**–**d**) Images obtained from a 4D-CT-scan, from maximal extension to maximal flexion, in a case of posttraumatic flexion–extension loss in a 56-year-old woman who suffered from a complex fracture–dislocation years ago. Posterior radial-head and capitellar osteophytes (white arrow) cause an extension bone block, whereas the anterior osteophytes (dotted white arrow) do not contact, suggesting a capsulous cause to the loss of flexion. Those osteophytes are better depicted on (**e**) a global illumination 3D reformat in extension and (**f**) flexion, also clarifying their anatomic relationships.

**Figure 11 jcm-10-05348-f011:**
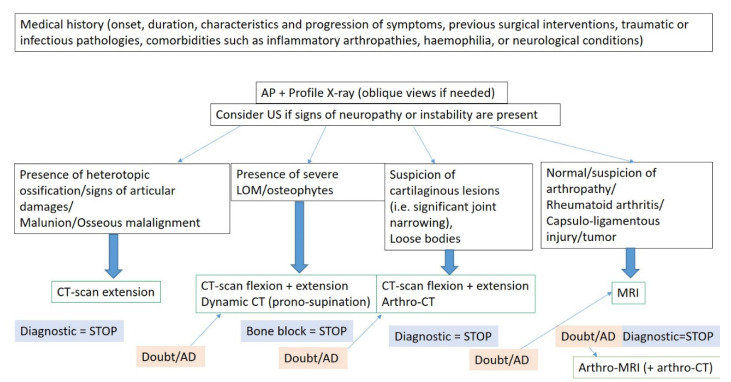
Diagnostic imaging studies prescription proposal. AD: alternative diagnosis; AP: anteroposterior; US: ultrasonography; LOM: loss of motion. The term doubt refers to an unsure diagnosis or a possible alternative diagnosis, requiring other imaging tests.

**Table 1 jcm-10-05348-t001:** Flexion–extension stiffness etiologies.

Extension Dysfunction	
** *Anterior Tether* **	** *Posterior Block* **
Thickened anterior capsule	Entrapped synovium
Contracture or HO of the anterior band of the medial collateral ligament	Osteophytes of the olecranon process or fossa
Contracture or HO of the flexor muscle/tendon	Loose bodies in the olecranon fossa
Anterior scarred skin	Posterior elbow ho
	Posterior articular osteochondral lesion
**Flexion Dysfunction**	
** *Posterior Tether* **	** *Anterior Block* **
Thickened posterior capsule	Entrapped synovium
Contracture or HO of the posterior band of the medial collateral ligament	Osteophytes of the coronoid process or fossa
Triceps contracture or HO	Loose bodies in the coronoid fossa
Posterior scarred skin	Anterior elbow ho
	Anterior articular osteochondral lesion

HO: heterotopic ossification.

**Table 2 jcm-10-05348-t002:** Radiographic checklist.

Items to be Checked on Radiographs	
*Anteroposterior view*	*Profile*
Malunion	Malunion
Malalignment in the frontal plane	Malalignment
Loose bodies	Loose bodies (olecranon and coronoid fossae)
Joint space narrowing	Radial head subluxation or dislocation
Periarticular calcification	Drop sign * (instability?)
Heterotopic ossification	Heterotopic ossification
Osteophytes (radial head)	Capsular ossification (anterior +++)
	Osteophytes (olecranon and coronoid +++)
	Olecranon, coronoid, or radial fossae osteophytic filling

*: The drop sign corresponds to an irregular humeroulnar joint space, superior to 3 mm in width [21].

**Table 3 jcm-10-05348-t003:** Ultrasonographic checklist.

Ulnar neuropathy +++
Heterotopic ossification vasculonervous relations, if present
Cartilaginous lesions
Joint effusion and synovitis
Loose bodies (and their mobility)
Ligamentous pathology in case of instability
Osteophytes’ location

**Table 4 jcm-10-05348-t004:** CT and CT-arthrography checklist.

**Loss of flexion**	**CT/CT-Arthrography**
	*Posterior tethers/anterior blocks*
	HO
	Anterior synovitis
	Loose bodies
	Osteophytes/fibrosis in the radial or coronoid fossa
	Osteophytes around the coronoid
	Joint space narrowing
**Loss of extension**	**CT/CT-Arthrography**
	*Posterior blocks/anterior tethers*
	HO
	Posterior synovitis
	Free bodies
	Osteophytes/fibrosis in the olecranon fossa
	Osteophytes around the olecranon
	Joint space narrowing
**Forearm rotation dysfunction**	**CT**
	Proximal radioulnar bony ankylosis
	Malunion or nonunion of the radial head
	Posttraumatic sequelae of the radial head
**Any stiffness**	**CT-Arthrography**
	Humeroulnar joint cartilaginous lesion (absent/mild/moderate/severe)
	Humeroradial joint cartilaginous lesion (absent/mild/moderate/severe)

HO: heterotopic ossification.

**Table 5 jcm-10-05348-t005:** MRI checklist.

Loss of Flexion
*Anterior tethers/posterior blocks*
Capsular scarring (lateral and medial ligament complex)
Brachialis scar
HO
Anterior synovitis (consider IV contrast media if needed)
Loose bodies
Osteophytes in the radial or coronoid fossa
Osteophytes around the coronoid
Joint space narrowing
**Loss of extension**
*Posterior blocks/anterior tethers*
Capsular scarring (lateral and medial ligament complex)
Triceps scar
HO
Posterior synovitis
Loose bodies
Osteophytes in the olecranon fossa
Osteophytes around the olecranon
Joint space narrowing
**Forearm rotation dysfunction**
*Not initially indicated*
Chondropathy of the radial head
Annular ligament scar
Radioulnar synostosis
**Any stiffness**
Humeroulnar joint impingement (absent/partial/severe)
Humeroradial joint impingement (absent/partial/severe)
Soft tissue lesion
Ulnar nerve injury

HO: heterotopic ossification.

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
