# Peer review of "Elbow Stiffness Imaging: A Practical Diagnostic and Pretherapeutic Approach"

_jcm, 2021, doi:10.3390/jcm10225348_

Round 1

Reviewer 1 Report

Dear Authors,

Thank you for the opportunity to review this manuscript. It is an interesting, thorough, and well-illustrated narrative review incorporating clinical experience.

Overview:

Well-constructed manuscript outlining recommendations for the pre-theraputic imaging workup of patients with elbow stiffness, encompassing all imaging modalities.

I encourage you to consider the addition of an ultrasound checklist table to complement the other modalities' tables, and some refinement of your final diagnostic algorithm flowchart, Figure 11.  You should also note in the Abstract that your review leads to a proposed diagnostic algorithm.

Other minor points you may find useful:

Page 1 of 15 –

line 17 ‘Imaging main concern is to determine the cause of elbow stiffness’ – please check grammar

line 23 – The elbow joint

page 2 of 15 –

line 71 - ‘Stiffness in flexion is less tolerated than in extension, as supination loss compared to pronation’ – please review.

line 78 –may lead to a flexion deficit

line 80 – associated with

Table 1 – a careful review of capitalised words is needed here

page 3 of 15

line 92 – described

line 113 - but also lead to a forearm

Line 115 –consider: and delays before surgery

Page 4 of 15 –figure 2: please review the legend for grammar and flow

Page 5 of 15

line 143 –inconsistent offset of subheading

Line 155 -158 –please check grammar and typos

Page 7 of 15 –

line 210 - you mentioned that CT is gradually replacing the complimentary radiographic views you described; consider providing your opinion on this

Page 8 of 15 –

Line 217 - provide a reference for the ‘drop sign’.

line 218 – please consider an ultrasound checklist to compliment your tables for the other modalities

line 234 to 236 –please review the sentence starting with ‘it also allows’

Line 244 –should this be ‘processes’ rather than ‘processing’?

Line 248 –consider 'may remain 'rather than 'will remains '

Page 9 of 15

line 266 –consider: have been shown to be good candidates for arthroscopic treatment

Line 267 –check grammar

Line 272 –consider ‘depicted’ rather than ‘depictable’

Figure 8 legend - please check spelling and grammar

Page 10 of 15 –

Table 3 –please consider an alternative table format which maps clinical indication against modality

line 285 – I do you mean ‘and’ capsular thickening?

Page 11 of 15 –

line 304 –Loose bodies

Figure 9 - please review figure legend for grammar

Page 12 of 15 –

line 314 - cinematic with a C – also in line 336

line 318 - joint blocks are frequent

Page 13 of 15 –please check for typographic and grammatical errors throughout including figure 11

Figure 11 – consider adjusting this figure such that the bottom row with multiple uses of ‘doubt’ more clearly indicates that the doubt is an alternate to diagnostic, bone block, etc

Page 14 or 15 –references –please check all references carefully

Author Response

Dear Reviewer, 

Thank you for the opportunity of improving our work.

A point by point response is attached.

Best regards

Reviewer 2 Report

I think this work summarizes and comprehensively explains a problem widely present in clinical practice.

The topic of the paper it is relevant for clinical practice because more often we cane see stiffness problems, often in athletes. The topic it is not original but it is original the the approach with which the topic is addressed is it: “ A Practical Diagnostic and Prethera peutic Approach” ;the reader can use this paper as a toolbox during his daily professional activity The paper is well written with some minor grammar error.

Author Response

Dear Reviewer,

Thank you for your positive comments. Our efforts have been rewarded.

Grammatical errors have been corrected.

Best regards.